# Imaging Techniques for Detecting Prokaryotic Viruses in Environmental Samples

**DOI:** 10.3390/v13112126

**Published:** 2021-10-21

**Authors:** Victoria Turzynski, Indra Monsees, Cristina Moraru, Alexander J. Probst

**Affiliations:** 1Department of Chemistry, Environmental Microbiology and Biotechnology (EMB), University of Duisburg-Essen, Universitätsstraße 5, 45141 Essen, Germany; indra.monsees@uni-due.de; 2Institute for Chemistry and Biology of the Marine Environment (ICBM), Carl-von-Ossietzky-University Oldenburg, Carl-von-Ossietzky-Straße 9-11, 26111 Oldenburg, Germany; liliana.cristina.moraru@uni-oldenburg.de; 3Centre of Water and Environmental Research (ZWU), University of Duisburg-Essen, Universitätsstraße 5, 45141 Essen, Germany

**Keywords:** fluorescence microscopy, electron microscopy, helium-ion microscopy, atomic force microscopy, metagenomics, viromics, fluorescence in situ hybridization, virusFISH, phageFISH, direct-geneFISH

## Abstract

Viruses are the most abundant biological entities on Earth with an estimate of 10^31^ viral particles across all ecosystems. Prokaryotic viruses—bacteriophages and archaeal viruses—influence global biogeochemical cycles by shaping microbial communities through predation, through the effect of horizontal gene transfer on the host genome evolution, and through manipulating the host cellular metabolism. Imaging techniques have played an important role in understanding the biology and lifestyle of prokaryotic viruses. Specifically, structure-resolving microscopy methods, for example, transmission electron microscopy, are commonly used for understanding viral morphology, ultrastructure, and host interaction. These methods have been applied mostly to cultivated phage–host pairs. However, recent advances in environmental genomics have demonstrated that the majority of viruses remain uncultivated, and thus microscopically uncharacterized. Although light- and structure-resolving microscopy of viruses from environmental samples is possible, quite often the link between the visualization and the genomic information of uncultivated prokaryotic viruses is missing. In this minireview, we summarize the current state of the art of imaging techniques available for characterizing viruses in environmental samples and discuss potential links between viral imaging and environmental genomics for shedding light on the morphology of uncultivated viruses and their lifestyles in Earth’s ecosystems.

## 1. Introduction

In the last few decades, it has been proven that viruses represent the most abundant components in Earth’s ecosystems [1]. Based on the molecular composition of their genome, these viral predators can be categorized into DNA and RNA viruses, whereas the respective genome can be single-stranded (ss) or double-stranded (ds). Current estimates based on aquatic and soil ecosystems suggest that the majority of prokaryotic viruses have dsDNA genomes [2]. By influencing microbial communities on various levels, viruses impact carbon, nitrogen, sulfur, and phosphorous cycling in many ecosystems. For instance, up to 10^28^ viral infections are estimated to occur in the ocean per day [3], which convert cells into dissolved organic matter (DOM) and particulate organic matter (POM) [3], leading to a viral shunt and a redistribution of carbon compounds in the stratified ocean [4,5]. Beyond the killing-the-winner hypothesis, based on which viruses modify host diversity [5] and microbial community composition, the piggy-back-the-winner hypothesis was recently proposed, highlighting the importance of viral lysogeny in microbial communities [6]. Viruses alter biogeochemical cycling of microbes via horizontal gene transfer. Moreover, viruses can introduce auxiliary metabolic genes (AMGs) [7] to their prokaryotic host, adjusting the metabolism to their needs and likely increasing their burst size. Recent metagenomic studies have demonstrated that AMGs are diverse and abundant in marine viromes [8,9,10] but can also occur in archaeal and bacterial viruses in the deep subsurface [11]. For instance, viruses can heavily impact sulfur cycling in the deep ocean by transferring genes for sulfate reduction [8]. Furthermore, the presence of photosystem I genes in marine cyanophages results in a metabolism boost of Prochlorococcus and Synechococcus cells during viral reproduction [12]. While these findings were first documented using metagenomic studies [13], the isolation of such viruses containing genes for photosystem I and II was successfully reported only eight years later, highlighting the necessity of studying uncultivated viruses directly in ecosystems. The Tara Ocean Project [14] was set out with the aim of investigating microbial interactions, their evolution, and viral infections [15] in the sea ecosystem. This project was based on an extraordinarily large dataset containing millions of newly discovered sequences from various oceanic microbes and viruses [15], and—along with the Pacific Ocean Virome (POV) [16,17] project and the Malaspina expedition [18]—revolutionized our view of genetic diversity of prokaryotic and eukaryotic viruses on Earth. Such datasets represent the best current means of documenting the taxonomic compositions of uncultivated and unknown viral communities [16] in various ecosystems.

In contrast to uncultivated viruses, those that were cultivated in the laboratory have been extensively studied for decades, including via imaging techniques [19], and have substantially broadened our knowledge regarding viral morphologies and ultrastructure. Fundamental insights regarding how viruses interact with their hosts and regarding the discovery of novel viruses can be gained from using epifluorescence microscopy, electron microscopy (EM), helium-ion microscopy (HIM), and atomic force microscopy (AFM) (Figure 1, Table 1). For instance, virus enumeration (determined as virus-like particles, VLPs) is routinely performed via epifluorescence microscopy and EM for several ecosystems, e.g., VLPs range from 10^7^ to 10^8^ mL^−1^ in marine and freshwater environments [4,20,21] and from 10^8^ to 10^9^ VLPs cm^−3^ in sediment [22,23] and soil [24,25]. In addition, the combination of both imaging techniques (fluorescence and electron microscopy) —known as Correlative Light and Electron Microscopy “CLEM”—can be combined for the quantification of viruses and the visualization of virus–host associations [26]. The benefit of this technical linkage is that the observation of fluorescent-labeled viral particles can be easily identified and tracked as “real viruses” by using EM [27]. For studying the viral entry and egress mechanisms, scanning electron microscopy (SEM) was identified as one of the most suitable imaging techniques [28]. Analyses based on cryo-EM have provided important information not only on investigating and reconstructing the viral structure [19,29,30] but also on the genome injection mechanisms of viruses [31]. Additionally, HIM and AFM have been applied for studying virus–host interactions under high resolution [32,33].

**Figure 1 viruses-13-02126-f001:**
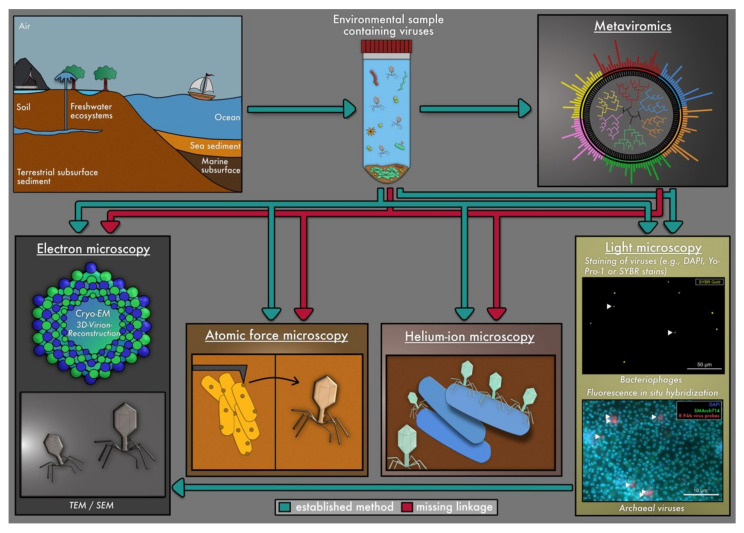
Overview of methods for detecting viruses in environmental samples and combinations thereof. Green arrows indicate commonly applied methods, which are described in the main text. Red arrows suggest potential couplings of methods that have not been performed to date. Coupling metaviromics with light microscopy has been performed in three studies (Rahlff et al., 2021 [10], Hochstein et al., 2016 [34], Jahn et. al., 2021 [26]) and is elucidated further in Figure 2. The origin of arrows in the figure is always associated with the environmental sample and the metaviromics analysis to indicate that both need to be combined to link viral structures to genome sequences.

**Figure 2 viruses-13-02126-f002:**
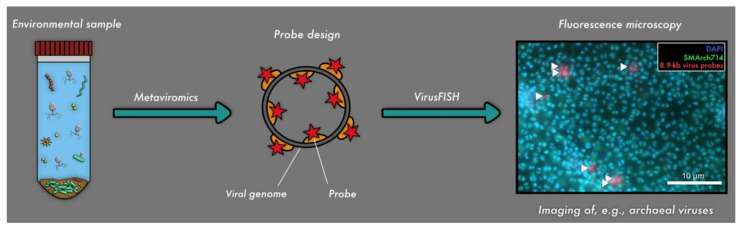
Linkage of metaviromics and fluorescence microscopy for detecting uncultivated viruses. The genome of the virus is detected in silico and used for respective probe design (Barrero-Canosa and Moraru, 2019 [35]), followed by detection of the respective viral genomes in situ (i.e., virusFISH [10]).

**Table 1 viruses-13-02126-t001:** Overview of frequently used microscopy methods for studying viral abundance, viral morphology, and viral diversity.

Microscopy Technique	Advantages	Disadvantages	Resolution	Coupling with Following Techniques Has Been Performed
**Fluorescence microscopy**	- Powerful, cheap, and simple technique [36] for analyzing viruses with high accuracy and precision [37]	- Photobleaching - No structural resolution, only detection	~300 nm for conventional light microscopy techniques and 10 nm for super-resolution microscopy (SRM) [38]	- Flow cytometry [39] - Ultracentrifugation [40] - TEM (as CLEM) [26]
Nucleic acid staining	- Estimation of total viral counts (more efficient than techniques such as TEM [22,41]) and can be carried out in the field [37]	- Some dyes have a long staining time or interfere with fixatives [42] - Small/large bacteria may be counted as viral particles [22] or bacterial cells [43] - Free nucleic acids can be recognized as viruses [44]		
Fluorescence in situ hybridization (FISH) (direct-geneFISH [45], phageFISH [46], virusFISH [10])	- Precise localization of gene signals and virus-infected/non-infected cells - Visualization of viral infections, e.g., from an early infection stage to viral bursts [46,47] and also used as culture-independent method	- Requires an experienced and trained person		- Metagenomics/viromics [10,26,34]
**Electron microscopy (EM)**	- Highest resolution - High-quality images	- Expensive equipment and time-consuming- Not possible in the field [48]	- Subnanometer resolution [38]	- Ultracentrifugation [49,50,51]
TEM, SEM and Cryo-EM	- Total viral counts, viral morphological characterization, viral infection frequencies, and burst size estimates can be obtained	- Underestimation of viral abundances [37,41,52]		
**Helium-ion microscopy (HIM)**	- Imaging of various stages of viral infections - Coating and embedding of the sample are not required (uncoated samples show more ultra-structures) [32]	- Helium is a limited element on Earth, expensive imaging technique	- Higher imaging resolution than techniques such as EM [32] - Nanoscale imaging capacity with a higher depth of view compared to AFM [32]	
**Atomic force microscopy (AFM)**	- Inexpensive method with mechanically and electronically straightforward instruments [53] - Staining, labeling, and coating the samples is not necessary [54] - Analysis can be performed in fluids and in air [53] - Obtaining of viral morphology	- Recording an image is time-consuming in contrast to fluorescence microscopy [53] - Limitation in scan range [53] - Debris can adhere to the AFM tip and can affect the image quality [53]	- Nanometer-scale resolution: imaging from molecular to cellular scale [33]	- X-ray diffraction [53] - Ultracentrifugation [55] - - HIM [56]

The difficulty of culturing the majority of host organisms poses a huge problem for the analyses of their viruses [57], and thus, the majority of viruses remains uncultivated and yet to be explored [3,24,58]. To overcome the challenge of linking uncultivated microbial hosts to their uncultivated viruses, in-silico techniques like metaviromics have swiftly become keys in analyzing viral genomes from various ecosystems and linking them to prokaryotic genomes. Although metaviromics has been used with different microscopy techniques, such as epifluorescence microscopy [10,34] and EM [10,34] (Figure 1), a direct linkage of genome and microscopic image is seldomly achieved [10,26,34] (Figure 2). Consequently, a major gap of knowledge exists regarding the morphology of uncultivated viruses, the stages of infection of their hosts, virus/host quantification, and the spatial localization of viruses and their host in ecosystems (e.g., in terms of interactions with other prokaryotic/eukaryotic microbes or in biofilms). The major bottleneck for studying these uncultivated viruses is the disconnection of sequencing data (metagenomics and viromics) from uncultivated viruses and microscopy techniques. In this review, we summarize the current state of the art of visualization techniques for (cultivated) viruses and discuss their potential linkage to sequencing data in order to explore viruses directly in ecosystems.

## 2. Retrieving Viral Fractions from Ecosystems for Microscopy Analyses

Quite often, viruses in environmental samples need to be concentrated and purified before being analyzed with microscopy techniques, e.g., for virus enumeration or identification of viral morphology. For this purpose, a careful removal of all contaminants like cellular organisms and debris is essential in order to obtain highly purified virus particles. Initially, viruses can be separated from contaminants, most often by size fractionation via filtration, but also via centrifugation (Table 1). The concentration and collection of viral particles can generally be performed either via tangential flow filtration (TFF), dead-end filtration or by using filters with pore sizes <0.02 µm, e.g., Anodisc™ or Nuclepore™ [37,59]. Analyses with Anodisc™ membranes provide an order of magnitude higher VLP abundance for seawater than Nuclepore™ membranes [59] and are thus likely better for collecting viral particles.

Concentrating viruses from environmental samples can be performed via TFF, in which the sample flows tangential to the filter surface [60]. Viruses from oligotrophic marine samples have been investigated using this concentration procedure in conjunction with ultrafiltration and transmission electron microscopy (TEM) analysis [61]. However, TFF can cause a loss in viral yield [62]. An alternative method for concentrating and purifying viruses from pure cultures and environmental samples is ultracentrifugation. Ultracentrifugation has been used on a range of viruses [40] and is conducted at more than 100.000× *g*, a centrifugal force that cannot be achieved by ordinary centrifuges. Thus, this technique requires an expensive ultracentrifuge, rotors, and special tubes [63].

Recent protocols combine ultracentrifugation with a flocculation, filtration, and resuspension (FFR) method in order to most efficiently concentrate viruses from environmental samples [64]. For instance, an iron-based virus FFR method showed a marine virus recovery of >94 ± 1% by using FeCl_3_, which represents an efficient, inexpensive, and non-toxic flocculant [64]. This flocculation method is beneficial because it allows large amounts of water to be processed and is technically simple and fast with high viral recovery [36,64].

Ultracentrifugation can also be used in combination with polyethylene glycol (PEG). Usually, a PEG concentration of 10% (*w/v*) is commonly used for virus precipitation [65]. When combining PEG precipitation with density gradient ultracentrifugation, a gradient media is required. This can be iodixanol [66] (“Optiprep™”), cesium sulfate, sucrose, or cesium chloride (CsCl) [49,63], all easily removed after centrifugation by adding dialysis steps. CsCl represents the most widely used gradient media for achieving highly pure and concentrated virus solutions [40]. For instance, insight into different viral morphotypes of uncultivated viruses infecting members of the genus *Acidianus* in an enrichment culture obtained from a volcanic area in Italy was achieved by coupling PEG and CsCl density gradient ultracentrifugation with TEM [50]. By using this method combination, *Acidianus* filamentous virus 1 (AFV1) was purified from an infected *Acidianus hospitalis* enrichment culture originally retrieved from hot springs [51]. These types of methods have been used so far in pure cultures and enrichments. The purification method based on the combination of PEG, CsCl density ultracentrifugation, and TEM has—to the best of our knowledge—not been applied directly to environmental samples.

## 3. Using Electron Microscopy for Virus Quantification and for Discovery of Previously Unknown Viral Morphologies and Ultrastructures

### 3.1. Sample Preparation for Transmission Electron Microscopy

Around the early 1930s, the first electron microscope was developed, and only eight years later the first electron micrograph of a virus was recorded in the literature [67]. Since then, TEM has been an integral part of the study of viruses for the determination of virion morphology and ultrastructure, as well as for virus quantification. For these purposes, two staining techniques are available—negative and positive staining [68].

Prior to the staining procedure, the sample is frequently concentrated through (ultra-) centrifugation and filtration by using 0.2 µm-pore-size filters to get rid of debris and cellular organisms (see above) [37]. Negative stains, phosphotungstic acid, ammonium molybdate, and uranyl acetate (also a fixative) are commonly used for staining the sample/phage solution deposited directly on the copper grid, whereby uranyl acetate is also used as a positive stain besides lead citrate for enhancing the contrast of samples [69]. The concentrated sample can also be embedded in epoxide or another resin. Thin sectioning of the embedded sample is performed by using a diamond knife to reveal the structures inside viruses and their hosts [70,71].

As a result of the two staining techniques, the negative staining depicts light viral particles on a dark background, while positive staining results in dark particles on a bright background. For both staining techniques, the samples are incubated in heavy metal salt solutions (e.g., uranyl acetate [72]), whereby the salt can react with cellular structures [72] or penetrate the viral tail [73], challenging the detection of fine viral structures. This paragraph just summarizes the superficial sample preparation for TEM that uses electron beams for the imaging of viruses, but correct preparation is essential for a thorough detection of viruses and elucidation of their morphologies. For further reading on viral preparation for TEM, the reader is referred to [74,75].

### 3.2. Estimating Viral Abundances in Environmental Samples Using TEMs

The first quantification of viruses in environmental samples resulted in the detection of approximately 10 million putative viruses per milliliter of seawater [76] and was conducted using TEM in 1989. Later, determining viral abundances via TEM was applied not only to seawater [77] but also to the slime of diatoms [78], marine snow particles [79], hydrothermal vent systems [80], deep granitic [81] and shallow aquifers [82], and marine subsurface sediments [23], resulting in a more general understanding of the distribution of viruses across Earth’s biomes. Quantifying viruses based on their capsid morphology was conducted by combining TEM with metaviromics across six oceans and seas through the Tara Oceans Expedition [83]. The aforementioned study resulted in the detection of four different viral morphotypes (*myoviral*, *podoviral*, *siphoviral*, and non-tailed icosahedral viruses), which were also analyzed on a genomic basis [83]. These analyses provided evidence that non-tailed viruses dominate the upper water column of the global oceans, comprising 51–92% of the observed viral particles [83]. TEM-based analyses of viral fractions did not only increase our understanding of the distribution of viral communities across ecosystems but also within ecosystems. For example, an increase in the viral abundance with depth was determined for marine sediment samples [23], a trend that was also later confirmed for archaeal viruses in deep granitic groundwater [81].

### 3.3. Determination of the Frequency of Visibly Infected Cells, Burst Sizes, and Spatial Distribution of Viruses

While the aforementioned studies focus on the detection of viral particles within the extracellular space, TEM can also be used for studying viruses within host cells. The frequency of visibly infected cells (FVIC) can be calculated for a sample by determining the number of visibly infected cells divided by the number of examined cells. FVIC is helpful in terms of estimating viral infections within cells, quantifying the number of infected cells, and determining the burst size (number of viral particles per cell). FVIC was calculated for several ecosystems, e.g., for the surface of glaciers (Arctic cryoconite) [84], various freshwater [85,86], and marine ecosystems [87]. Although this method only detects cells in the late stage of infection [88], it substantially increased our knowledge of naturally occurring virocells. For example, in marine ecosystems, the infection frequencies, burst size, capsid sizes, and the distribution patterns of viruses inside cells can vary between different bacterial morphotypes [87]. Moreover, coccoid and rod-shaped cells appeared to be more infected by viruses than, e.g., spirilla, in marine ecosystems [87], and likely also in alkaline lakes [89]. Prokaryotic mortality was also shown to vary among seasons (pre-monsoon, monsoon, and post-monsoon) with an increase in mortality during the monsoon season, whereas rod-shaped bacteria were more infected during the dry season [86]. Across all seasons, myoviruses were the most dominant viral morphotype besides non-tailed viruses, miphoviruses, and podoviruses [86]. The results of this study led to the general assumption that seasonal dynamics of viruses impact the carbon and energy flow in tropical esturine ecosystems [86].

### 3.4. Observing (Novel) Viral Morphologies

Viral morphology is one of the most studied viral features, usually by using TEM, because it offers information about how viruses are able to attach and penetrate the host cell. Furthermore, for a long time, morphological features determined by TEM, for example, tail type, capsid shape and size, and the presence or absence of envelopes and spikes [90], have been the main criteria used for viral classification, and, to a certain extent, they are still used today, together with the genomic information. For instance, until recently, the order *Caudovirales* (dsDNA viruses) comprised tailed viruses infecting bacteria and archaea, which were classified into three families (*Myoviridae*, *Siphoviridae*, and *Podoviridae*), based on their tail morphology [91]. Moreover, some viral particles have unique morphologies, such as lemon-shaped viruses [29,92], and some viruses undergo spectacular extracellular developments [50]. Most of these unique morphologies are associated with viruses that infect archaea. The biggest milestone for the study of archaeal viruses was the observation of the viruses in environmental samples via TEM in 1994 [93]. This analysis was quickly expanded to viruses infecting hyperthermophilic archaea in hot springs in the Yellowstone National Park (Wyoming, USA) [51,94]. One study particularly focused on one specific virus (*Acidianus* filamentous virus 1, AFV1 [51]), which infects only some representatives of the genus *Acidianus*. The authors revealed virus–host associations via EM pictures, which showed how AFV1 particles attach to the pili of *Acidianus hospitalis* and thus enable linking hosts and viruses in environmental samples [51]. Beyond the study of archaea-virus interactions with specific species in ecosystems, various morphotypes of hot spring viruses, ranging from filamentous, rod-shaped viruses to spindle-shaped viruses, were observed [94]. Ample research on hot spring viruses revealed a lemon-shaped *Acidianus* two-tailed virus with an astonishing extracellular morphological development [50]. This virus develops long filamentous tails at its ends after being released from its host cell if the virus is exposed to the correct ecosystem temperature of ~75 °C [50].

### 3.5. Scanning Electron Microscopy for Studying Unique Viral Egress Mechanisms

A special viral egress mechanism of archaeal viruses was revealed using a combination of TEM and scanning electron microscopy (SEM) by focusing on their morphology during their proliferation [28]. When exiting the *Sulfolobus* cell, the rod-shaped virus 2 (SIRV2 [95]) caused the formation of virus-associated pyramids [28] (known as VAPs) on the surface of the host cell after 10 hours of infection. During this last stage of infection, huge apertures were created for releasing the newly matured viruses [28]. Furthermore, the viral genome encoded the proteins that control the formation of VAPs [28]. TEM analyses also showed a change of the host phenotype during infection, highlighted by the absence of its S-layer [28]. The development of VAPs as virus release mechanisms was recently expanded to *Sulfolobus* turreted icosahedral virus (STIV) [96], suggesting that this complex exit mechanism is spread in the genus *Sulfolobus*, if not in other archaea as well.

### 3.6. Illustrating Virus–Host Associations by Using Cryo-Electron (Cryo-EM) Microscopy and Cryo-Electron Tomography (Cryo-ET)

Over the last 15 years, cryo-EM was established as the method to use when investigating in-depth and high-resolution viral structures [29,30,97] or virus–host interactions [98,99]. Cryo-EM is ideally suited for exploring the three-dimensional architecture of viruses at molecular resolution down to sub-nanometer resolution [100] (see Table 1 for a comparison with other EM techniques). For instance, cryo-EM enabled researchers to resolve the three-dimensional structure of the SIRV2 virion, revealing that the capsid protein wraps around the A-form viral DNA, to protect it [30]. Moreover, cryo-EM revealed that the *Acidianus* two-tailed virus (ATV) has two types of virion structures—tail-less and two-tailed virions [29]. The hyperthermophilic *Aeropyrum* coil-shaped virus (ACV) infects *Aeropyrum pernix* isolated from a coastal hot spring in Japan and has a single-stranded DNA genome [97]. TEM and cryo-TEM revealed that its virion is a rigid cylinder, a structure that has never been reported before for archaeal viruses [97].

Parallel to cryo-EM, cryo-electron tomography (cryo-ET) has recently emerged as a powerful tool in the study of viruses and can also be used like cryo-ET for illustrating virus–host associations. Both techniques are central in terms of determining the high-resolution structures of several viral assemblies [101]. Combining cryo-ET with subtomogram averaging methods can provide detailed 3D information of the structure of molecules, viruses, and their viral proteins at sub-nanometer resolution [101,102,103].

Recent advances in cryo-ET revealed various virus–host interactions and states of viral infections for bacteriophage Epsilon 15 and its hosts *Salmonella* [104], Phi29 and *B. subtilis* [105], T4 and T7 bacteriophages and *E.coli* [106,107], and for the cyanophage P-SSP7 and *Prochlorococcus marinus* [108]. Furthermore, there is a large volume of published studies describing the role of cryo-ET in investigating archaeal viruses. For instance, the first 3D structure of a *Sulfolobus* spindle-shaped virus (SSV1) was achieved with cryo-ET by Stedman et al. (2015) [109]. Only one year later, the complete viral entry and egress mechanism of this specific virus SSV1 was illustrated by using this technique, showing that these mechanisms occur on the cellular cytoplasmic membrane of the host *Sulfolobus shibatae* B12 [110].

A structural analysis of the filaments of *Acidianus* bottle-shaped virions (ABV) [50], the investigations of the tails of AFV3 virions [111], and the structure of the spindle-shaped virus His1 and their structural proteins were analyzed using cryo-ET and subtomogram averaging [112], showing the broad range of this technique. Another study on archaeal viruses revealed the VAP structure (further described in Section 3.5) by using subtomogram averaging [113]. The authors prepared sections by using the Tokuyasu method [114] and used antibodies for immunolabeling against protein-forming virus-associated pyramids (PVAP) for TEM analysis [113]. This study was the first that used the Tokuyasu method for TEM sample preparation on prokaryotic viruses, a method that uses a mild chemically fixation, freezing in sucrose and cutting the sample for cryo-sections. Despite the fact that the Tokuyasu method is not widely used in the area of prokaryotic viruses, this method has great potential when combined with CLEM for imaging and 3D structural determination of eukaryotic viruses as well [115].

Overall, electron microscopy has been used for three decades to study the morphology of viruses from environmental samples. It has proven to be a valuable technique that enables researchers to study not only the viral abundance in samples but also the details of viral proliferation, overcoming the limitations of light and fluorescence microscopy (Table 1). However, EM analyses are not only time-consuming and require more expensive consumables compared to light microscopy, but the microscopes themselves and their maintenance are costly. Furthermore, TEM analyses were shown to underestimate viral abundances due to uneven collection, problems during staining, washing off viral particles, and the generally low detection limit of the method [37,44,52]. In addition, TEM is not suitable for high throughput sample analysis like light and fluorescence microscopy. Despite these disadvantages, this imaging technique has helped to gradually expand our understanding of viral morphology and virus–host interactions and has contributed significantly to the discovery of many previously unknown viruses, particularly in complex environmental samples.

## 4. Shedding Light on Viral Abundances in Ecosystems Using Epifluorescence Microscopy

Fluorescence microscopy is often used as a first step to investigate viruses in a given sample. It represents a powerful, cheap, and simple technique [36] that enables researchers to quantify viruses (Table 1). This technique advanced the field of viral ecology since the former standard assay of determining viral loads via plaque assays [116] required a cultivated host and known culture conditions of the respective viruses [36]. Consequently, fluorescence microscopy is now the most frequently used technique for enumerating VLPs not only in pure cultures but also in environmental samples.

The first pioneer studies for determining the viral abundance in aquatic ecosystems via light microscopy were conducted in 1991 [52] and 1998 [37]. Since then, scientists have applied different fluorescence dyes, such as 4′,6-Diamidin-2-phenylindol (DAPI) [41,42,52], for enumerating viral particles, each with different advantages and disadvantages. For instance, T3 and T4 phage lysates and virus-sized particles in a high molecular weight concentrate (HMWC) of seawater were stained by using DAPI [117]. Because DAPI has low sensitivity when staining viruses, it was quickly substituted with Yo-Pro-1 [48]. Yo-Pro-1, a cyanine-based nucleic acid stain, is more stable and the fluorescence yield and binding coefficient for nucleic acids are higher [42,48]. Furthermore, this dye stains both DNA and RNA [20]. When the viral abundance in seawater was determined by using both dyes, DAPI and Yo-Pro-1, similar estimates were obtained [41]. Another study reported that Yo-Pro-1 and light microscopy is the “best” method for enumerating viruses [47]. Nevertheless, Yo-Pro-1 interferes with aldehyde-based fixatives [45]; therefore, it was eventually replaced by SYBR stains [37].

In 1998, a SYBR stain (SYBR Green I) was tested on marine water samples (containing viruses and bacteria) and was found to be an efficient dye for the field and for lab cultures [37]. SYBR Green I can be used for enumerating viruses, whereby a brighter fluorescence for viruses compared to stained bacteria can be achieved [37]. Later on, it was also applied to the quantification of micro-algal viruses, for example, the lytic virus PpV-01 infecting pure cultures of *Phaeocystis* [39]. This enumeration method was also applied to environmental samples from different oceanic locations (e.g., the English Channel, the Equatorial Pacific, and the Mediterranean Sea) [39]. Ample research using this fluorescent dye followed, for instance, for marine viruses from high turbidity seawater [98] and marine sediment samples [22,23], often combined with flow-cytometry [39,118] or confocal laser scanning microscopy (CLSM) [21]. CLSM was applied to aggregates from the Danube River (Austria), in which 5.39 × 10^9^ viruses cm^−3^ were found [21]. The detection of aggregate-associated viruses was performed with SYBR Green I [21], a stain that is frequently chosen for estimating viral abundances besides SYBR Gold [118]. SYBR Gold is more sensitive than SYBR Green I for staining viruses [118]. For example, viruses of surface marine and freshwater environments were investigated by using SYBR Gold [118]. Comparisons of virus counts were performed between conventional fluorescence microscopy and those determined with CLSM, and no significant difference between the methods was obtained [119]. However, CLSM is not currently well established for enumerating viruses [119], compared to the commonly applied epifluorescence microscopy.

Fluorescence microscopy also has disadvantages; for example, the detection sensitivity can be low in terms of the dyes [117] and over-estimations in viral counts due to artifacts [120]. For instance, the presence of gene transfer agents of membrane vesicles can lead to overestimating the viral load in a sample [121]. Moreover, small bacteria may be counted as viral particles [22]; in turn, large viruses are confounded with microbial cells [43], and not every fluorescent signal corresponds to a real virus (e.g., minerals can cause similar signals) [20]. Overestimation, as well as underestimation, can interfere with viral quantification. For instance, a recent study reported on detection issues of ssDNA viruses when using DNA-binding stains and fluorescence microscopy, attributable to the genome type (ssDNA) and small size [122]. Besides all these disadvantages, fluorescence microscopy offers an advantage in terms of speed and costs for sample preparation and analysis, as compared to EM.

To sum up, the studies mentioned above demonstrate that enumerating viruses by linking SYBR stains [37] with fluorescence microscopy is currently the most suitable method for estimating viral abundance in ecosystems. However, the actual confirmation of a VLP and the virus morphology cannot be studied in detail with light microscopy; thus, it requires a structure-resolving imaging technique like TEM, atomic force microscopy (AFM) or helium-ion microscopy (HIM).

## 5. Enhanced Surface Topography of Virus–Host Interactions Using Helium-Ion Microscopy

Helium-ion microscopy (HIM), an emerging technology, can be considered an ultra-high resolution “scanning” microscopy that uses helium atoms to scan the surface of a given sample [123]. Similarly to SEM, it can be used to analyze the topography of biological structures [32]. Sample preparation for HIM is simple, requiring no staining or embedding, which is a great advantage compared to conventional (S)EM. Furthermore, HIM has the potential to be used for investigating virus–host interactions [32] in terms of viral adsorption to the host cell and virus egression (Table 1).

To date, there is only one study that has investigated the association between viruses and their hosts in pure cultures by using HIM [32]. The first step in the field of bacteriophages was the observation of the interaction of *Escherichia coli* and T4 phage by using HIM [32]. Imaging of bacteria–phage interactions is difficult in terms of resolution limits, sample preparation, and the complexity of currently used microscopy techniques [32]. However, HIM enables researchers to image microcolonies growing on agar plates, thus presenting a new opportunity in the field of imaging [32]. For instance, active and individual infections of cells in plaques on agar plates were illustrated using this technique [32]. In addition, the tailed morphology of T4 phages with an icosahedral shape of the capsid head and attaching phages on *E. coli* cells [32] and their complete lysis stage were observed. As a side note, bacteria–bacteria interactions (without phages) can also be illustrated with HIM, as shown for the model organism *Flavobacterium columnare* B185 [32].

HIM application in environmental samples has been so far limited to a single study [124] (Figure 1). In the aforementioned study, sediments and microbial mats from Himalayan hot springs were investigated using HIM and SEM to reveal the presence of tailed and non-tailed icosahedral viruses, and spindle-shaped viruses. This morphological assessment was supported by viral diversity data inferred from metagenomics.

Overall, HIM has great potential for deciphering viral structures and has the potential to slowly become a key instrument in the analyses of not only virus–host interactions but also microbe–microbe interactions. However, helium is a very limited element on our planet [125], and extensive use of HIM might further shorten its availability.

## 6. Atomic Force Microscopy for Cost-Effective Scanning of Viral Structures

Atomic force microscopy (AFM), invented by Binning et al. in 1986 [126], enables the visualization of the structural characteristics of viruses without pretreating the sample, similar to HIM. AFM integrates several imaging techniques [54], e.g., scanning ion-conductance microscopy (SICM [127]), scanning electrochemical microscopy (SECM [128]), Kelvin force microscopy (KFM [129]), and scanning near-field ultrasound holography (SNFUH [130]). Given this array of techniques, AFM also has several disadvantages, which include, for instance, the extended time for recording an image compared to fluorescence microscopy (see Table 1). Furthermore, cell remnants and other debris can adhere on the AFM tip (as contamination) and may affect the quality of the overall image [53]. Other disadvantages comprise forces acting between probe and sample, which can affect the image resolution [131]. Nevertheless, AFM currently represents the only technique that can image the surface of living cells at nanometer-scale and in real-time [131]. Moreover, AFM can also be combined (like EM) with X-ray diffraction, whereby a crystallized virus for the direct visualization of viral structures is required [53]. A recent study demonstrated the potential of AFM by integrating this technique into a helium-ion microscope [56]. Until now, no study on viruses has yet been conducted by using this powerful technical linkage, which could help characterize novel viruses in various ecosystems.

Using AFM, researchers investigated a variety of different viruses, e.g., plant viruses (turnip yellow mosaic virus, TYMV [132], satellite tobacco mosaic virus [133]), algal virus *Paramecium bursaria Chlorella* virus 1 (PBCV-1 [134]), and the largest known virus, *Acanthamoeba polyphaga Mimivirus* [135,136]. One study focused on important protocols for the investigation of different bacteria–phage interactions by using AFM [33]. This pioneering study includes protocols that span diverse host–phage systems including *E. coli* 057 and its lytic bacteriophage A157, *Salmonella enteritidis* 89 and its lytic phage 39, and *Bacillus thuringiensis* 393 and its V_f_ phage [33]. The abovementioned study visualized the ultrastructure of phage particles and compared bacterial surfaces in different phases of infection via AFM [33]. Besides imaging the outer surface of bacteria in order to detect viral infections, the inside features of, e.g., broken and intact viral capsids of bacteriophage ΦKZ [137], as well as lysed bacteriophage capsids, e.g., from phage T4 [138] due to osmotic shock or surface attachment, were also visualized by using AFM.

Phage–host interactions were also characterized by AFM in direct comparison with TEM, whereby more intact phages were found via AFM compared to TEM [33], due to the fact that AFM provides a three-dimensional format of the image. In addition, phage DNA [33] can also be visualized via AFM, which was also reported for PBCV-1 [134] and for phage RNA of several icosahedral viruses [139].

These examples demonstrate that extensive research with AFM has been carried out on phages grown under laboratory conditions; however, no study has—to the best of our knowledge—used AFM to investigate viruses of environmental samples so far.

To summarize, AFM is an inexpensive method (little maintenance, little costs for consumables compared to EM, a fairly cheap machine) for studying the presence and morphology of viruses [140]. In general, AFM has different modes (contact mode, non-contact mode, and tapping mode), making it an ideal tool for studying living microbial systems in air, liquid, and solid samples [53]. In general, AFM cannot replace the functions and information that can be obtained by light/fluorescence microscopy and EM [141], as it is not as efficient as light/fluorescence microscopy, nor is its resolution comparable to EM or HIM. However, AFM provides a cheap and easy-to-use method for obtaining information on the viral morphology or illustrating the viral surface without complicated sample preparation.

## 7. Virus Discovery by (Meta)genomics and Microscopy

The majority of microorganisms cannot be cultured under defined laboratory conditions [142,143]. Therefore, it is not surprising that the majority of viruses have not been isolated so far, because this requires cultivated hosts [57]. Studying the genetic inventory of complex microbial and viral communities can be accomplished by metagenomics, particularly if this technique is used to resolve genomes of populations from the ecosystem [124,144]. Although we demonstrated above that there exists a plethora of literature on studying viruses in environmental samples using microscopy, only a limited number of studies applied both environmental genomics of viral communities and respective microscopy techniques [124,145]. In the following section, we discuss studies that performed metaviromics and microscopy separately, and which did not directly link the virus genome sequence and the viral morphology from environmental samples.

A recent study focusing on the hottest terrestrial geothermal spring in South Africa applied the combination of fluorescence microscopy, electron microscopy, and metagenomics [146]. EM analysis on 74 VLPs revealed tailed bacteriophages of the order *Caudovirales* [146]. The metagenome of this spring included a highly abundant cyanophage genome and a genome fragment of a virus likely infecting *Gemmata* [146]. However, the three methods were not linked; for instance, fluorescence microscopy was not applied on the exact same field of view as electron microscopy. Also, the visualization was not directly linked with genome information, which also applies to the following studies.

In another study, the viral diversity of Manikaran hot springs was investigated by using metagenomic profiling and SEM, revealing archaeal viruses with different morphotypes, mainly belonging to the *Fuselloviridae* family [124]. Four metagenomes from sediment samples of the hot spring were analyzed, by which 59 different archaeal viruses (including *Fuselloviridae, Siphoviridae* and *Podoviridae*) of varying abundances were identified. Two additional metagenomes of microbial mat samples aided in reconstructing another 65 bacteriophage genomes [124]. These studies highlight the general bottleneck in environmental viromics; viral morphology can provide only limited insight into virus taxonomic diversity [83], while the physiological characteristics of viruses can barely be extrapolated from their genetic sequences [147].

The first step in linking viral genomic information and morphological analyses from EM was recently performed in a study set out with the aim of investigating the microbial and viral diversity of five deep terrestrial subsurface locations (hydraulically fractured wells) [148]. *Halanaerobium* spp., the most common species in this ecosystem, was in situ actively predated by viruses, as revealed by studying infection histories based on the host’s CRISPR array [148]. Viral genomic data revealed 50 viruses with a putative lysogenic lifestyle and 3 integrated prophages identified in isolate genomes of *Halanaerobium* retrieved via cultivation [148]. The lysis of the cultivated *Halanaerobium* WG8 by its tail-less virus (burst size: 61 viruses per lytic event) was visualized by TEM [148], representing an indirect linkage of metagenomics and TEM via cultivation assays.

Taken together, studies that used metagenomics and microscopy demonstrate the great potential of these techniques regarding the analysis of the uncultivated majority of viruses. Furthermore, linking these two technologies, i.e., assigning infections or ultrastructure to genomic information, could further bolster our understanding of viral ecology; the abovementioned studies only realized this by cultivating the respective host.

## 8. A Promising Technique for Linking Environmental Genomics to Fluorescence Microscopy of Viruses

### 8.1. Fluorescence In Situ Hybridization (FISH) for Tracking Virus–Host Interactions

FISH uses genomic sequence information for probe design in order to identify biological entities in fluorescence microscopy [46]. At the same time, this method also enables the quantification of microorganisms in samples [149,150] (Figure 1). In microbial ecology, FISH is widely used for the identification of microorganisms based on the ribosomal RNA of the small subunit (16S rRNA) [151]. In addition, several other FISH methods targeting rRNA, mRNA, and DNA have been developed [45,46,152] (see Table 1).

Consequently, the idea of targeting viruses with oligonucleotide probes in FISH received much attention in recent years, as it would—in theory—enable researchers to visualize uncultivated viruses in their environment and determine their infection rate. A gap of knowledge relates to the linkage of genomic sequences to the morphology of uncultivated or previously unexplored viruses, the infection stages of their hosts, the viral lifestyle (lytic or lysogenic cycle), and virus enumeration in ecosystems. To perform such linkage, coupling FISH with TEM/HIM could help by filling these research gaps. However, the major bottleneck for studying viruses is the FISH protocol itself (e.g., the permeabilization of the cells) because the sample preparation tends to destroy the ultrastructure of microbial cells [153] and probably the viral morphology as well. Nevertheless, permeabilization is crucial in FISH since permeabilization of the cells allows the labeled oligonucleotides to diffuse through the cell envelope [154].

Nowadays, different FISH methods exist for detecting viruses, e.g., cycling primed in situ amplification-FISH (CPRINS-FISH) [155] was developed for detecting individual genes in a bacterial cell, which was then optimized to visualize viral DNA and phage-mediated gene transfer in freshwater environments [152]. More recently, scientists developed PhageFISH [46] as an optimized technique of GeneFISH [156]. PhageFISH enables the visualization of viral infections from the early stage of infection to bursting cells [46]. This type of method was further extended to eukaryotic microorganisms. For instance, the infection process of the eukaryotic algae *Ostreococcus lucimarinus* by Prasinoviruses was studied, whereby 200 infected cells with different stages of infection were counted [47]. In this study, virus-attached cells (viral signals detected on the margin of host signals), infected cells (the overlapping of virus and host signals), and lysed cells (concentrated viral signals and lost host signals) were observed [47]. More recently, PhageFISH and direct-geneFISH [45] were combined for applying virusFISH on the unicellular green algae *Ostreococcus tauri* and its virus *Ostreococcus tauri* 5 (OtV5) to monitor viral infections in pure cultures [88]. Furthermore, a new approach called Virocell-FISH [157] showed the interaction of a giant virus chasing its host *Emiliania huxleyi*, an alga that is part of the ocean’s biomass [155]. Consequently, there exists a great array of potential FISH methods for targeting viruses, whose potential has not yet been leveraged for exploring viral communities and their dynamics in ecosystems.

### 8.2. Coupling of Metaviromics with Fluorescence In Situ Hybridization

Although the knowledge gained through the variety of available microscopy techniques is astonishingly high regarding viruses in general, coupling meta-omics techniques with microscopy appears to be a major challenge and is consequently not frequently represented in the literature (see Figure 1). While some research has been carried out regarding metaviromics and microscopy separately (mentioned in the previous section), there have been few studies that coupled both techniques directly. Crucial information for the design of probes that detect viruses in FISH results from metaviromics and thus aids in designing the detection of viruses in, e.g., phageFISH [46]. The linkage of metaviromics with imaging techniques was first described and performed for an environmental sample by Hochstein et al. (2016) [34]. They identified a new archaeal *Acidianus* tailed spindle virus (ATSV, host: *Acidianus hospitalis* W1) with a 70.8 kb circular dsDNA viral genome in an acidic hot spring (Alice Spring) in the Yellowstone National Park (Wyoming, USA) [34]. They sequenced the viral genome and designed probes to identify the virus within the ecosystem and confirmed this after isolating the virus and visualizing the ATSV virus particles in TEM as large, spindle-shaped virions [34]. A comparative study also used genomic information to perform FISH targeting a viral genome in an enrichment culture [158]. An archaeal *Metallosphaera* turreted icosahedral virus (MTIV) [158] was isolated from an acidic hot spring in the Yellowstone National Park (USA), and its icosahedral morphology was illustrated via TEM, cryo-electron tomography, and single-particle analysis. In order to verify their virus–host prediction, direct viral FISH [45] was performed on *Metallosphaera yellowstonensis* cultures [158].

A recent investigation on the relationship of subsurface viruses targeting abundant primary producers (here Altiarchaeota) was conducted by Rahlff et al. (2021) [10]. Intracellular and extracellular localizations of various stages of viral infections in Altiarchaeota biofilms by using virusFISH [10], a modified version of direct-geneFISH [45], were detected (Figure 2). Super-resolution microscopy was also performed on the Altiarchaeota biofilms, resulting in a better resolution of the viral infections [10] and representing the first time a super-resolution microscopy was used for the study of virus-infected cells. Applying direct-geneFISH on multiple samples from the ecosystem, the authors showed that the genome sequence detected by metaviromics belongs to a lytic virus and thus challenged a standing paradigm that the subsurface is mostly populated by lysogenic viruses.

Another example of coupling virusFISH with metagenomics was represented by replicating a lysogenic phage in a marine sponge holobiont (here *Aplysina aerophoba*) [26]. In this study, a new in situ microscopy approach called ‘PhageFISH-CLEM’ was developed that is based on a combination of imaging and bioinformatics for investigating virus–host interactions, providing new insights into virus ecology within marine sponges [26]. Phage-FISH-CLEM results showed phages within bacterial symbiont cells and in phagocytotic active sponge cells, indicating that lysogeny dominated this sponge microbiome [26]. For the first time, phageFISH was coupled directly with EM and enabled researchers not only the quantification of viruses but also the visualization of virus–host associations originally predicted from metagenomic data.

## 9. Conclusions

Viruses are a central component of Earth’s biome and shape complex microbial communities and are crucial for ecosystem functioning [58]. Studying these tiny microbial predators in natural environments is necessary because so few of them can be cultivated under laboratory conditions. Over the last five decades, researchers have used microscopy to discover different viral lifestyles along with unique morphologies to demonstrate active viral infections across all domains of life. Research to date has consistently shown that viruses can be detected with various imaging techniques (epifluorescence microscopy, EM, CLEM, AFM, HIM). Applying metaviromics and microscopy, researchers have so far discovered only a small portion of the genetic and morphological diversity of viruses on our planet.

Tremendous advances have been made not only in the field of microscopy by developing higher resolution instruments, improved staining methods, and novel labeling strategies for FISH analyses, but also in the field of in-silico analyses of viromes for detecting sequences of novel viruses (VirSorter [159], VirFinder [160], Vibrant [161]). The huge sequence data sets obtained through metagenomics and viromics, which, in particular, facilitate the development of new FISH probes that are essential for studying more and previously unexplored viruses in environments [34,46]. Development of these technologies will in the future not only lead to the visualization of other novel viruses but could also lead to an improved understanding of the role viruses play in environmental communities, by, e.g., quantifying viral infections in environmental samples.

We propose that linking (meta)viromics and FISH to AFM, HIM, and EM, i.e., correlative light and electron microscopy (CLEM), for environmental samples will reveal the morphology of uncultivated viruses that have only been identified on a sequence basis so far. The plethora of different viral morphologies, particularly in the realm of archaeal viruses, promises many fascinating insights into viral diversity once CLEM becomes more applicable to ecosystems. For viruses that cannot be imaged via CLEM, there exists the possibility of identifying novel capsid proteins via proteomics of viral fractions followed by recombinant expression of the proteins and antibody generation for immunogold labeling in TEM [162]. The overview as well as the new perspectives presented in this study will hopefully aid other researchers in the exploration of the morphology of cultivated and yet-to-be cultivated viruses on our planet in order to better understand their diversity and functioning in Earth’s biomes.

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
