# Peer review of "Imaging Techniques for Detecting Prokaryotic Viruses in Environmental Samples"

_viruses, 2021, doi:10.3390/v13112126_

Round 1

Reviewer 1 Report

This is a nice review of current methods for prokaryotic virus visualization, I particularly appreciate the extensive references to archaeal virus studies.  Personally, I would have liked to see more emphasis on the Phage- and Virus-FISH analyses pioneered by this group, but that is not necessary.  I would also would have liked to see more future ideas on the linking of (meta)viromics to direct imaging techniques.  However, this is also not necessary.

General comments: Be sure to be consistent with “viromics” “metagenomics” and “metaviromics” throughout the manuscript.

Specific comments:

L31: I disagree that this is “recently”, maybe “relatively recently” or “in the last few decades”

L48: Is reference 9 really meant for groundwater ecosystems?  It seems a little out of place.

L66: I suggest citing a general review here or “Fields Virology”

L75-77: While I really like these references, there are also numerous bacteriophage and other viruses that could be referenced here, maybe a review or two instead of a couple of primary literature references.

L80: I found the Figure to be a little hard to follow at first, the arrows are a little confusing as the red arrows are a combination of viromics and environmental samples.  The green arrow on the right should also then be a combination of viromics and environmental samples, maybe it could be green and red, to indicate that this is a combination.  Or some way to indicate that viromics data is being used (or proposed to be used) in addition to the environmental sample.

L83-84: “Please note that” could be deleted (particularly if changes are made to the figure as suggested above.

L98: a connection to EM including viromics is not shown in Figure 1.

L115: Suggest adding “concentration and” before “collection”

L118: Suggest adding “Analysis with” before “Anodisc”

Table 1: Suggest “in the field” instead of “on field trips”

L124: Suggest deleting “along a parallel” and “directly”

L127: Suggest replacing “abundances” with “counts” or “yield”

L130: Suggest replacing “centrifuge speed” with “centrifugal force”

L133: If correct, suggest replacing “ultracentrifugation” with “centrifugation”

L163: Suggest using “and” instead of “or”.

L228: While ATV has a lemon-shape at one part of its replicative cycle, the Fuselloviruses are a much better example of lemon-shaped viruses.

L231: I would suggest instead of this reference ([72]) to use “Screening for Sulfolobales, their Plasmids and their Viruses in Icelandic Solfataras”, Zillig et al., System Appl. Microbiol. 16 609-628 (1994), which presents most of the data reviewed in reference 72.

L256 and following: This section could also include cryo-ET for virus-host interactions, including Quemin et al., 2016 (doi:10.1128/mBio.01439-16.) and Hu et al, 2013 (doi: 10.1126/science.1231887)

L307: Suggest “in the field” instead of “field trips”

L313: Suggest “using” instead of “on”

Reviewer 2 Report

The manuscript by Turzynski et al. reviews imaging techniques for detecting viruses in environmental samples.

The review is well-written and a nice summary of the techniques used in the field.

Major:

The authors should include a better introduction to explain the differences between techniques that bind samples directly onto the grid (e.g. negative stain) and sample embedding and subsequent thin-sectioning (lines 165f).

Moreover, a clearer distinction between single-particle reconstruction using cryo EM and cryo-electron tomography with subsequent subtomogram averaging should be made.

Also the term CLEM is only defined in the last paragraph and should be explained earlier. CLEM can also be used to make sure that DNA labels specifically detect virus particles in fluorescence microscopy. Also the use of DNAse/RNAse to degrade extraviral DNA before staining should be discussed.

Finally, I would suggest trying to do Tokuyasu sample preparation and embedding with subsequent thin-sectioning and infiltrating with biotinylated "FISH" probes and detecting with Streptavidin-gold 

Minor:

-the authors need to fix several reference errors (Reference source not found)

-Table 1: Electron microscopy: "highest magnification". This should read "resolution"
